# Unveiling LoRa’s Oceanic Reach: Assessing the Coverage of the Azores LoRaWAN Network from an Island

**DOI:** 10.3390/s23177394

**Published:** 2023-08-24

**Authors:** João Pinelo, André Dionísio Rocha, Miguel Arvana, João Gonçalves, Nuno Cota, Pedro Silva

**Affiliations:** 1Atlantic International Research Centre, 9700-702 Angra do Heroísmo, Portugal; pedro.silva@aircentre.org; 2NOVA School of Science and Technology, Center of Technology and Systems (UNINOVA-CTS) and Associated Lab of Intelligent Systems (LASI), NOVA University Lisbon, 2829-516 Lisbon, Portugal; andre.rocha@uninova.pt (A.D.R.); m.arvana@uninova.pt (M.A.); 3Independent Researcher, 9700-702 Angra do Heroísmo, Portugal; 4Instituto Superior de Engenharia de Lisboa, 1959-007 Lisboa, Portugal; nuno.cota@isel.pt

**Keywords:** maritime communications, LoRa, LoRaWAN, low power wide area network (LP-WAN), Internet of Things (IoT)

## Abstract

In maritime settings, effective communication between vessels and land infrastructure is crucial, but existing technologies often prove impractical for energy-sensitive IoT applications, like deploying sensors at sea. In this study, we explore the viability of a low-power, cost-effective wireless communication solution for maritime sensing data. Specifically, we conduct an experimental assessment of the Azorean Long Range Wide Area Network (LoRaWAN) coverage. Our tests involve positioning the gateway at the island’s highest point and installing end nodes on medium-sized fishing vessels. Through measurements of received signal strength indicator (RSSI), signal-to-noise ratio (SNR), and lines of sight (LOS), we showcase the potential of LoRaWAN transmissions to achieve communication distances exceeding 130 km in a LOS-free scenario over the ocean. These findings highlight the promising capabilities of LoRaWAN for reliable and long-range maritime communication of sensing data.

## 1. Introduction

The Internet of Things (IoT) can be described as a network of machines, sensors, actuators, and physical devices that can interact with each other and their environment, collecting information and transferring data to the network without any human interference, connecting the physical and digital worlds [1]. The IoT has proliferated in recent years, finding applications in various industries, such as energy, health, logistics, security, and agro-industry [2]. The maritime sector also exploits IoT [3], focusing mainly on logistics [2]. However, IoT applications can also be applied to maritime and marine activities such as aquaculture, offshore platforms, ocean exploration, fishing vessels and generic maritime monitoring [4,5,6]. Additionally, it may also be applied to fishing gear and touristic activities. In these contexts, reliable low-power machine-to-machine communication presents one of the biggest challenges.

Recently, significant advances have been made regarding wireless communications, specifically with the introduction of the sixth generation (6G). This type of communication, more specifically the fourth and fifth generation, 4G and 5G, respectively, has shown excellent results in terrestrial IoT applications. However, when the communication environment is maritime, there are many challenges [7] rooted in this type of environment, namely large distances with a low density of users. In the case of wireless communications (4G and 5G), base stations are not installed in the sea, which, allied to their relatively short radius of coverage, restricts their usage to onshore and coastal scenarios. In this case, communication cannot be established between vessels or vessels and land, and therefore, the application of IoT in the maritime/marine environments will remain limited to uses within a ship.

A set of technologies that are used or expected to be used in maritime communications was identified by [8,9]. Satellites like Argos, Iridium, or Iridium NEST are considered a pillar of maritime communications, mainly used for long-range communication scenarios. However, this technology is associated with high costs and bandwidth limitations. Most medium and short-range ship-to-ship or ship-to-shore communication technologies use the radio frequency (RF) band, more specifically, medium frequency (MF), high frequency (HF), very high frequency (VHF), and ultra-high frequency (UHF) bands. However, although these technologies can reach long propagation distances, they only provide small data rates, limiting their applicability to basic scenarios. In addition to using the RF band, the infrared band is also used. This, together with optical sensors, creates a solution known as free space optics (FSO), capable of providing connectivity to maritime communication networks. However, the FSO signals can be affected by weather and sea conditions, eventually creating variations in the optical signals or the plane of the receiver (vessel). Mobile communications (GSM) or WiMAX technology are also used in maritime scenarios. Recently, a solution based on unmanned aerial vehicles (UAVs) was proposed to improve the coverage of terrestrial-satellite communication networks [10]. Although the technologies mentioned above are used in maritime communications, their power requirements can only be met with access to an electric grid or a powerful energy source. With these technologies, the power needed for the communications module is often several orders of magnitude larger than that of the sensors. Less energy-intensive communications would allow sensing the environment more autonomously and for longer periods. For example, sensors could measure and communicate for long periods with a small battery.

Lately, Low Power Wide Networks (LPWANs) have been widely used in monitoring applications in several areas. LoRa is one of the main LPWAN communication technologies, and its use has been studied for maritime applications, ranging from monitoring fishing vessels [5] and passengers [8] to water quality monitoring in fish farms and coastal areas [6,11]. LoRa technology provides large coverage distances, low deployment costs, and low energy consumption [12]. However, it also has limitations such as low bandwidth, high vulnerability to interference, and low transmission rate [13].

### 1.1. Research Questions

Starting with a scenario of oceanic communications, we tested the viability of establishing a communication network with wide coverage and low cost in the context of the Azores archipelago. The network aims at real-time machine-to-machine communications between small fishing vessels (including ones without a battery bank) and the islands, primarily transmitting timestamped positioning data.

Considering that traditional maritime communication technologies cannot comply with the energy restrictions that some systems face, as well as the final purpose of the network, the following research questions are posed:–RQ1: Which wireless communication solution enables real-time data extraction on small vessels with limited power?

**Hypothesis** **1 (H1).**
*Employing LoRaWAN will facilitate the communication of long-range, low-bandwidth data between small vessels and gateways on an island.*


This may lead to the second question:–RQ2: At what maximum distance can effective LoRaWAN communications be carried out between the vessels and the gateway for tracking their location?

**Hypothesis** **2 (H2).**
*By deploying a node (IoT device) on a vessel and installing a gateway on land, it is feasible to establish data communication between the two entities for a distance of approximately 100 km.*


### 1.2. Document Structure

This paper is structured as follows. The next section presents a literature review on LoRa and LoRaWAN. This is followed by the materials and methods section, and subsequently by the results and discussion. Lastly, concluding remarks are provided in the final section. The main contributions of this work are:A series of measurements on the deployed network which focus on signal quality and maximum distance of communication.Despite packet loss, we establish that communicating the vessel position at five-minute intervals is sufficient to map the vessel location for practical use cases.We investigate the impact of line of sight (LOS) vs. non line of sight (NLOS).We gained traction to develop the network further and perform more robust testing.

## 2. LoRa and LoRaWAN

Long Range (LoRa) low-power networks are becoming increasingly popular [14]. These networks are an effective communication solution owing to their stability and low cost, and because they do not require licensing, they present a low barrier to implementation. LoRa also has low power and low bandwidth. While this would be a limitation for human communications, it is adequate for machine-to-machine communication, particularly sensors, whose usage is growing exponentially. The low power of LoRa is also critical for the use case of sensors, which, due to low power demand, do not need to be tied to the grid, operating from batteries, frequently combined with photovoltaic cells.

The low entry barriers of LoRa, namely the low cost associated with the freedom for implementation, differentiate LoRa networks. Unlike other communication solutions, the economics of LoRa allows for the distribution of gateways as necessary for adequate coverage, even where there is a lower density of use due to the lower viability (cost-benefit) threshold.

LoRa is one of the leading Low Power Local Area Networks (LPWAN) communication technologies being adapted worldwide to connect sensors [15], forming the Internet of Things (IoT). In the IoT, resources tend to be scarce. IoT devices need less memory, processing power, and energy requirements, allowing for ubiquitous sensing, which is unfeasible based on other communication technologies such as Wi-Fi or traditional cellular networks [1,12].

The LoRaWAN network architecture (illustrated in Figure 1) is based on a star-of-stars topology, with three types of devices: end device (node) (typically one or more sensors with a LoRa communication element), LoRa gateway (LoRa communication device that receives messages from the end devices), and a LoRa network server, which aggregates the data obtained through the gateways.

The communication between the end devices and the gateways is LoRa-based. LoRa is composed of two layers: one physical and one logical. The physical layer, LoRa, is Semtech’s proprietary wireless modulation technique, which uses chirp spread modeling [13]. The logical layer, LoRaWAN, is a Media Access Control (MAC) layer protocol built on top of the LoRa (physical) modeling layer, designed primarily for sensor networks. It defines how devices use the LoRa hardware, e.g., when they transmit and the format of messages [13]. The communication between the gateways and the network server tends to use higher bandwidth backhauls such as 3–4–5G or ethernet. As shown in Figure 1, the LoRaWAN network architecture allows for bidirectional communication. In the case of end devices, communication is performed with a relatively low frequency (e.g., hourly, or daily, not lower than at two-minute intervals). It is the responsibility of the network server to provide the acknowledgements and select the best links to forward the downlink messages [16].

**Figure 1 sensors-23-07394-f001:**
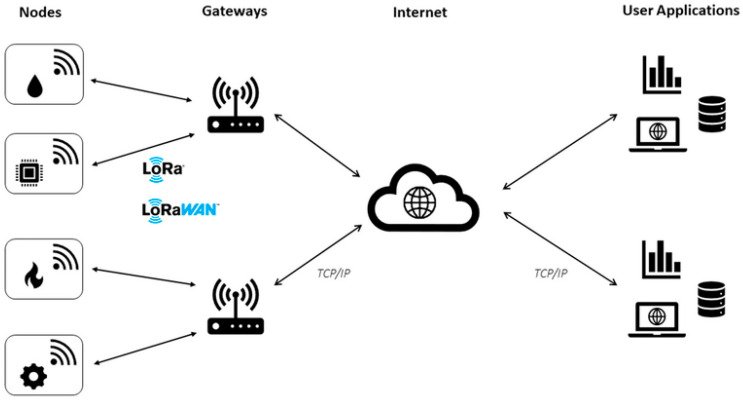
LoRaWAN network architecture [17].

LoRa is well documented in the literature [12,18]. While the physics of LoRa transmission seems well suited for communication over the ocean, which is naturally free of permanent obstacles, the truth is that there are not many trials and tests where these characteristics have been tested and validated [19,20,21]. And to our knowledge, no trials have been conducted on an archipelago.

We set up a LoRaWAN network for Terceira Island in the Azores, Portugal. It covers over 90% of the island and all targeted areas. Since, with the current use, we are far from consuming the available duty cycle, we aim at using it for different applications over the ocean. If reliable service could be extended to approximately 100 km from the shore, then the usability of the network would increase by several orders of magnitude.

Compared to the existing literature, our test results encourage and augment the potential for using the network to easily cover use cases previously unfeasible or complex and expensive to support, such as small fishing vessels or fishing buoys.

## 3. Materials and Methods

### 3.1. Equipment

The experimental setup included a LoRaWAN network composed of a gateway, a network server, and an end device equipped with GPS tracking. Wanesy Management Center, a cloud-based SaaS provided by Kerlink (Torigne Fuillard, France), was used as the network server and the device management platform.

The gateway consisted of a Kerlink Wirnet iStation (Figure 2a), a LoRaWAN EU868 certified Outdoor Gateway with 8 RX channels of 125 kHz and multi Spreading Factor, 1 RX channel of 500 kHz and mono Spreading Factor, and 1 RX channel with FSK modulation, for a total of 10 RX channels with a sensitivity of −141 dBm. It also has 1 TX channel configured with a power of 14 dBm, an SoC composed of a Cortex A9 ARM CPU and 256 MB of DDRAM, 8 GB eMMC flash storage, and the backhaul is secured by worldwide 4G module with 3G/2G and Ethernet (RJ45) fallback.

The gateway is coupled to a Kerlink ACCIOT-KAN01 antenna (Figure 2b) with vertical polarization, 865 MHz +/− 5 MHz frequency range, 50 Ohms impedance, and a max gain of 6 dBi. Antennas with lower gain (1–2 dBi) are used more frequently, but since the goal was to assess the distance covered on the sea, we used an antenna with higher gain that is still free of license and does not require a special power source.

The end device (Figure 2c) consisted of a Yabby from Digital Matter, a small battery-powered device that is LoRaWAN EU868 certified with GPS tracking. The device has a uBlox EVA-M8 GPS receiver with 72 channels and −167 dBm sensitivity. It also includes a 3-axis accelerometer for motion detection. The device was configured with a downlink spreading factor (SF) 12.

**Figure 2 sensors-23-07394-f002:**
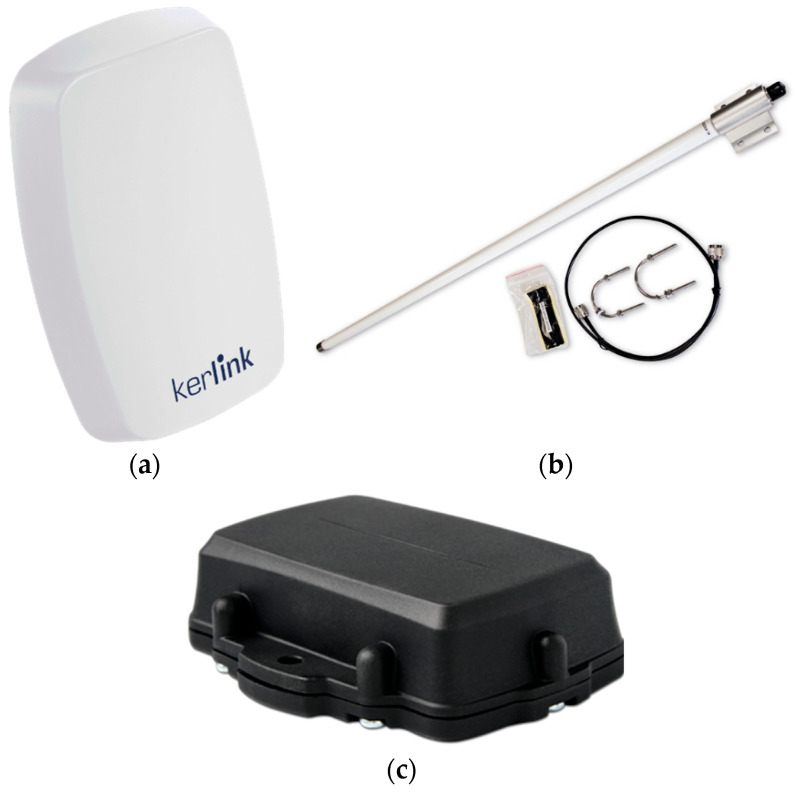
(**a**) Physical appearance of Kerlink Wirnet iStation [22]; (**b**) Kerlink ACCIOT-KAN01 antenna [23]; (**c**) Yabby from Digital Matter [24].

### 3.2. Scenario

At the time of writing, four LoRaWAN gateways are installed on Terceira Island, with the setup described above covering approximately 90% of the island’s territory. Table 1 shows the model, location, and altitude of each gateway. This, in turn, is far from being at maximum capacity.

In the first step, we tried to understand if the service provided on land could be reliably extended to at least 100 km from the coast to track fishing vessels. To test the viability of the service over the ocean, the gateway located in the Serra de Santa Barbara was selected. Mapping the aggregated density of vessels over long periods can be helpful for the management of large Marine Protected Areas (MPAs) or the Exclusive Economic Zone (EEZ). This initial assessment investigates the potential to support use cases that are not critical and can handle incomplete data due to packet loss and other phenomena. Depending on the potential demonstrated in future tests, different use cases that extend to critical might emerge. However, those must be based on a better knowledge of network performance, including packet loss and the impact of weather conditions, among others.

The choice of the gateway was based solely on the height above sea level. Figure 3 shows the location of the Serra de Santa Barbara gateway on Terceira Island, as well as the terrain elevation of the entire island.

The Azores is an archipelago composed of nine islands, five of which are relatively close to each other. This makes communication impossible in some cases because the islands themselves can act as barriers to communication, as shown in Figure 4, where the theoretical coverage of the gateway is represented at a maximum distance of 150 km.

### 3.3. Experiment

For this experiment, the end device was installed on a fishing vessel, and the location was tracked based on GPS (Global Positioning System) location transmitted at five-minute intervals (when in motion). Considering regular vessel speeds around 5 knots, it takes a vessel 5 min to move by a distance of 750 m. Since vessels tend to follow relatively straight lines, occasional missing packets do not imply a significant difference between a logic spatial inference and the actual route, rendering the output adequate for mapping aggregated activity zones across long periods. The gateway receives the location data from the end device as a LoRa message and forwards it to the network server. As it arrives at the network server, it is pushed to a REST Webhook developed to handle the LoRa messages from the end device along with the battery voltage, RSSI and SNR network-level metrics. The tracking device (hardware) did not support logging the packets sent; therefore, tracking packet loss was impossible at this stage.

The process of handling payloads from the GPS tracking device involves several steps. The device sends a message containing encoded location information to a gateway (via LoRaWAN), which then forwards it to a network server (via the internet). Upon receipt, the server pushes the news and received signal strength indicator (RSSI) and signal-to-noise ratio (SNR) metrics to a webhook. The Webhook was responsible for the decoding of the LoRa payload containing the location and other data and for creating new entries in a table of a database with the following columns: timestamp; device latitude; device longitude; gateway id; RSSI; SNR and battery voltage. Gateway locations are stored in another table in the database and contain each gateway’s id, latitude, longitude, and elevation. The webhook was built using Node and Express.js, and the payload decoding was processed using a JS function provided by the device manufacturer. It accepts POST requests to the uplink endpoint, ‘/up’. To ensure secure data transmission, the connection to the webhook is established via SSL (Secure Sockets Layer). Additionally, only incoming requests with a valid API key are accepted, ensuring that only authorized parties can access the data and preventing unauthorized access or tampering. The decoded location information and signal metrics are then stored in an online MySQL database for easy access. The payload handling process described above ensures secure and efficient data transmission from the LoRaWAN device to the network server and database.

## 4. Results and Discussion

In this section, we present the results of the tests that aimed to assess the LoRaWAN service over the ocean. Figure 5 represents the 12,381 points (dataset) collected between January and December 2022. These points come from devices associated with vessels with the same frequency of location sampling. Figure 5 illustrates different routes the vessels travelled, where each point was communicated from the vessel, along with signal characteristics. The positions recorded are coherent with the trips reported. Some routes show a discontinuity. These may be associated with an area of variable coverage. Later, we will further analyze these discontinuities, considering radio wave patterns and terrain elevation profiles.

The superposition of Figure 5 with the map of the Azores archipelago resulted in a heat map of all readings taken, as shown in Figure 6. Figure 6 better illustrates the areas mostly used by the vessels and the distances of successful transmissions obtained between the gateway and the devices (nodes).

Sub-GHz frequencies (868 MHz Europe) minimize signal attenuation due to obstacles and provide a robust modulation, allowing receivers with very low sensitivities of −140 dBm. In the case of our implementation, the device used for signal reception has a sensitivity of −141 dBm. Therefore, for analysis purposes, −140 dBm was the minimum Received Signal Strength Indicator (RSSI) power value considered to obtain a reliable transmission. However, according to the literature, a signal is considered weak for RSSI <= −120 dB. For the Signal-to-Noise Ratio (SNR), which is the ratio between the received signal power and the noise power level, the values of considered SNR in LoRa networks are between −20 dB and +10 dB [20,25].

Exploring the dataset, we analyzed the relationship between the collected data, namely the RSSI, the SNR, and the distance between the sender and the receiver. Figure 7 illustrates the relationship between RSSI and the distance covered by the signals (distance between the sender (node) and receiver (gateway)). There is an expected decrease in signal strength with increased distance between the sender and receiver. Shorter distances also have low RSSI values ([0.40] km); we investigate this below.

Figure 8 illustrates how the SNR varies as a function of the distance between the sender and receiver. One may observe that the events are within the limits of LoRa transmissions, and most have a positive value, many close to +10 dB. This indicates that, generally, the received signal operates above the noise floor, which suggests quality LoRaWAN transmissions across an extensive range of distances. However, disparate SNR values are also observed, which may be associated with areas of no network coverage (shadowed areas), obstacles in the line of sight (LOS), or too large a distance between the sender and receiver.

Figure 9 illustrates the relationship between SNR levels to RSSI and suggests possible origins of some disparate values. This analysis can be seen in Figure 9. Most readings are in the range [RSSI >= −120 dB and SNR >= −7 dB]. This indicates that the GW is in a good location, and the quality of the received signals is also good. For the readings that are in the ranges [RSSI >= −120 dB and SNR < −7 dB], [RSSI < −120 dB], there may be factors that have influenced these readings to disperse from the optimal zone. The scatter can be affected by the high noise floor, the transmitter being too far from the GW, or an obstruction at the LOS, ultimately causing packet loss or non-decoding of the signal.

Data subsets were used in a more detailed analysis to clarify some of the disparate values. Subsets were created considering the range of distances to the gateway and the density of points within a 3 km radius, i.e., for different distances (from close to far), the locations with the most samples were chosen to form a subset. The application of the criteria mentioned above gave rise to the subsets that can be seen in Figure 10.

The subsets are described in Table 2.

Figure 11 plots the average RSSI value of each subset against its average distance to the gateway. At first glance, there are unusual RSSI values from the 0 to 40 Km distances, specifically regarding the orange, pink, and yellow subsets. However, these subsets contain a mix of high and low values. Also, there are subsets within the areas in question with normal/expected RSSI values. An analysis of the line-of-sight (LOS) between the sender and the receiver reveals the possible cause for the disparity of values within subsets.

Figure 12, Figure 13 and Figure 14 illustrate the LOS analysis conducted for the orange, pink, and yellow subsets, respectively. The occurrence of LOS obstruction by the terrain is verified for the orange and pink subsets. Besides the losses associated with free space path loss, which are expected, events such as LOS obstruction cause attenuations due to terrain shielding. In a simulated environment, the attenuations are 32 dB for the orange subset and 11 dB for the pink subset. The obstruction of the LOS explains the unexpected RSSI values for these two subsets. Still referring to the LOS analysis, although the terrain does not block the LOS, the yellow subgroup invades the Fresnel zone. The Fresnel zone is an imaginary zone with a relative radius between the distance from the transmitter to the receiver and the transmitting frequency. The Fresnel zone assumes an elliptical shape along the LOS between the transmitter and receiver. As a rule, this zone should always be unobstructed; however, this is not always the case, so it is said that after 40% blockage of the site, there will be significant losses in the signal. In the case of subset yellow, there is a blockage greater than 40%, which, in a simulated environment, manifests itself in a signal attenuation of an order of 4 dB.

For the remaining Red, Green, Blue, Purple, Brown, Gray, Black, White, and Cyan subsets, all present LOS and Fresnel zones without obstacles. The RSSI values of these subsets are considered adequate relative to their distance from the GW. They are also consistent with the high sensitivity associated with LoRa technology, even if they are very close to the recommended limit of −120 dB.

In the analysis performed on the SNR values, the Green, Blue, Yellow, Orange, Purple, Pink, Brown, and Black subsets are representative of a noiseless transmission channel, with matters within limits previously defined and those associated with LoRa technology. For the Red, Gray, White, and Cyan subsets, the SNR levels, although within the defined limits, represent a noisy transmission channel.

Analyzing the subsets’ RSSI and SNR value pairs according to [21], reveals that the Blue, Orange, Purple, and Brown subsets are in the spectrum where the radio frequency level is optimal for reliable reception. The Red, Green, Yellow, Pink, and Black subsets are in the range where the RF levels are not optimal but sufficient. Transmissions have difficulty reaching the GW and improved link quality would imply reducing the distance to the gateway, while guaranteeing free LOS. The Gray, White, and Cyan subsets, on the other hand, are in the spectrum where the transmitter is too far from the receiver, as can be seen from the “Average Distance to Gw” values in Table 2.

We used the GPS positions of the vessels collected during the tests to understand the range of a LoRa and validate the simulated coverage area (Figure 4). This allowed us to calculate the distance between the end device installed on the vessels and the GW on the island. The results show that it is possible to communicate at distances greater than 130 km between the end devices and the GW if unimpeded by obstructions. However, further testing is needed to verify the reliability of these communications.

## 5. Conclusions

This work describes the results of assessing the ocean coverage of a LoRaWAN network on an Azorean Island. The main goal of this study was to verify the feasibility of establishing a communication network with a wide range, low cost, and based on the IoT concept for the whole archipelago. For this purpose, a set of tests was performed using the existing infrastructure, specifically the gateway of Serra de Santa Barbara, located at the highest point of Terceira Island.

To validate the coverage, a theoretical and experimental study was performed. For the experimental validation, end devices were placed on a small fishing vessel to test the effectiveness and range of radio transitions between the open ocean and land for distances greater than 100 km. The results were promising, evidencing the feasibility of receiving LoRa transmissions considering end devices installed on mobile platforms moving at sea in coastal and offshore scenarios and in free LOS and NLOS (non-line of sight) conditions. However, in NLOS conditions, shadow zones causing signal attenuation were found, as predicted in the theoretical study. In free LOS conditions, the results were in line with the theoretical research, and reliable transmissions were achieved at distances greater than 130 km between the end device and GW.

Despite validating the theoretical coverage, the set of tests performed is necessary to standardize and stabilize the network coverage. Therefore, there is still work to be done in this direction. As future work considers the stabilization, uniformity of network coverage, and minimization of shadow areas, such as those that were verified, it is proposed to implement new GWs. Developing a new test panel including new metrics, such as packet loss, is suggested for this. Packet loss was not tested due to the hardware limitations of the nodes. Further analysis will provide greater details regarding the coverage and help identify ideal locations for new GWs.

## Figures and Tables

**Figure 3 sensors-23-07394-f003:**
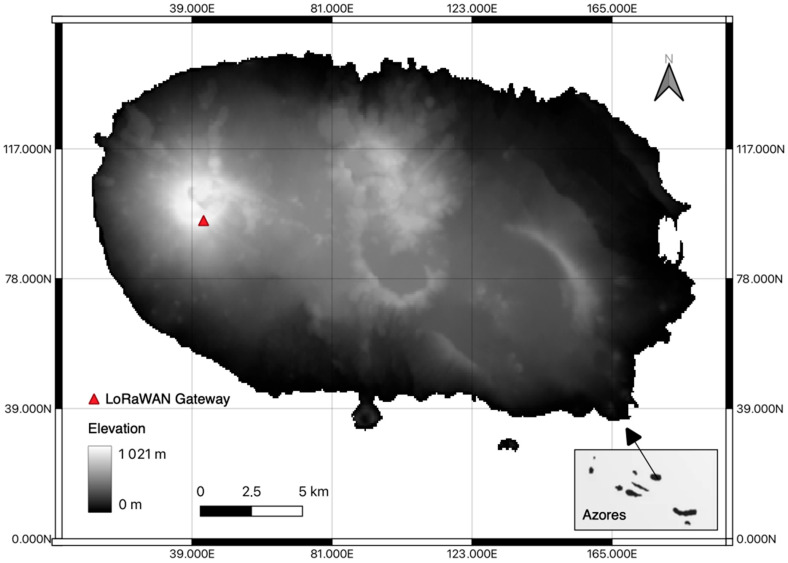
Location of the Serra de Santa Bárbara gateway.

**Figure 4 sensors-23-07394-f004:**
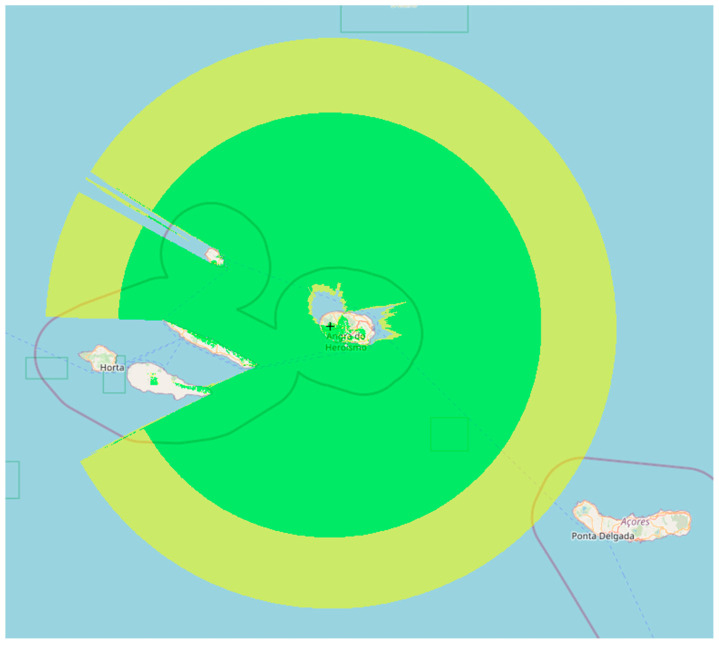
Theoretical coverage (150 km) of the Serra de Santa Barbara gateway.

**Figure 5 sensors-23-07394-f005:**
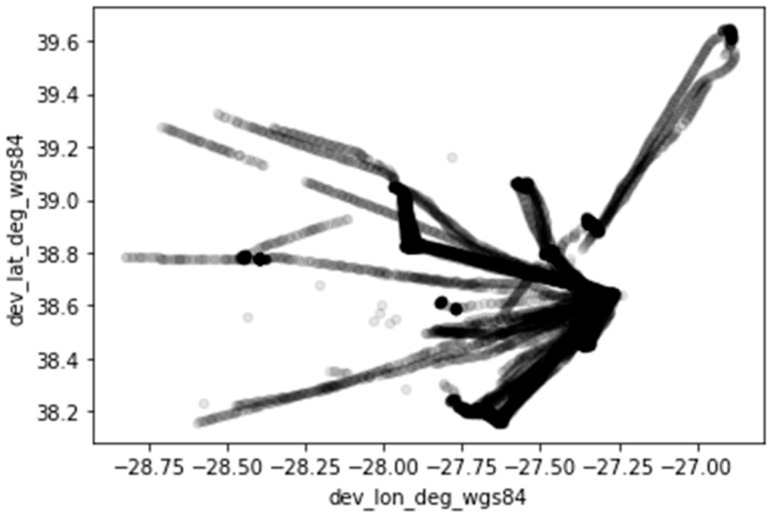
Location of readings taken between January and December 2022.

**Figure 6 sensors-23-07394-f006:**
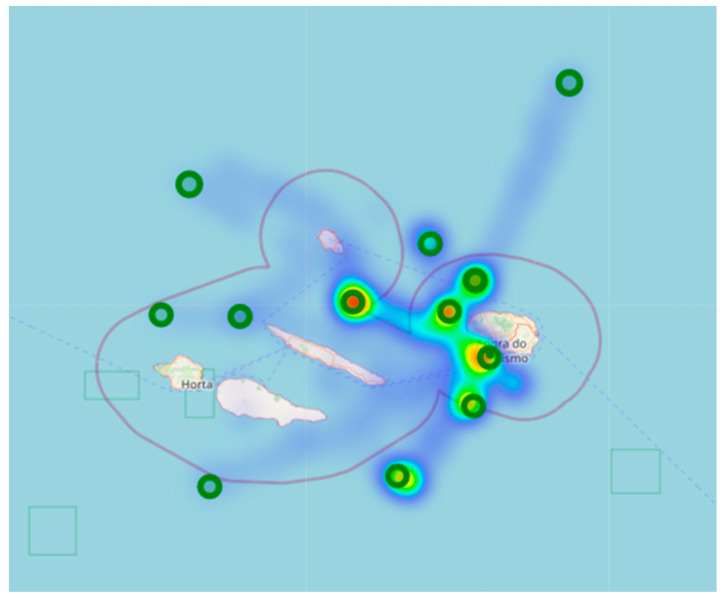
Heat map of location readings.

**Figure 7 sensors-23-07394-f007:**
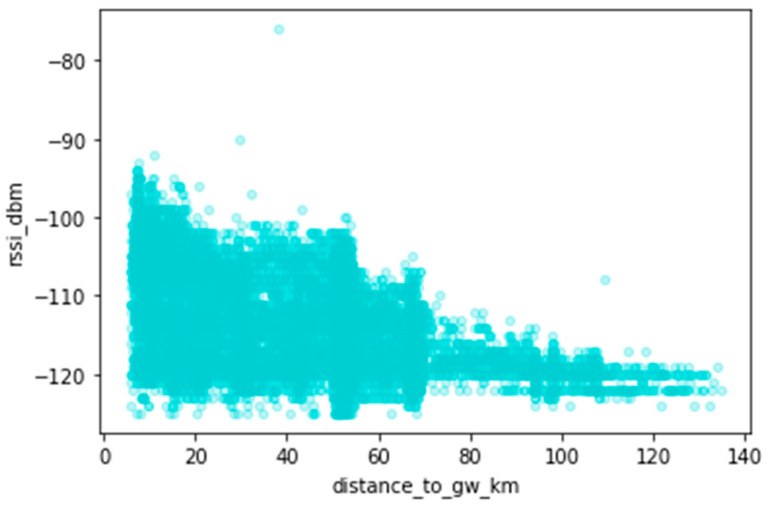
Received Signal Strength Indicator (RSSI) as a function of distance.

**Figure 8 sensors-23-07394-f008:**
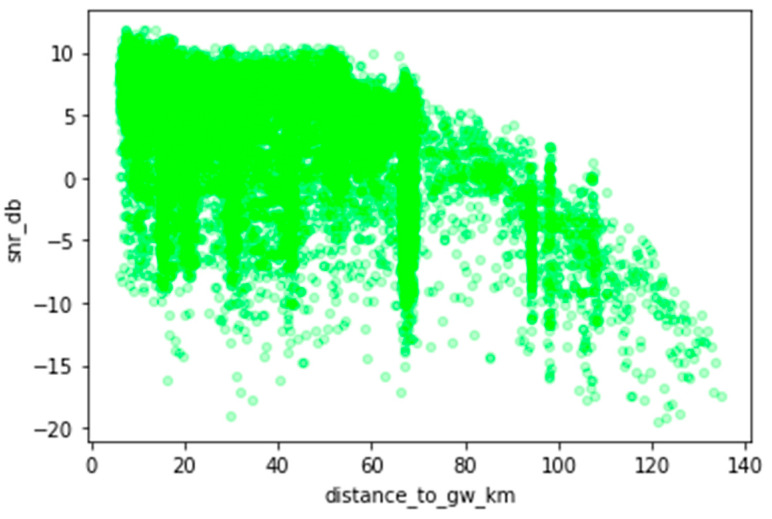
Signal-to-Noise Ratio (SNR) as a function of distance.

**Figure 9 sensors-23-07394-f009:**
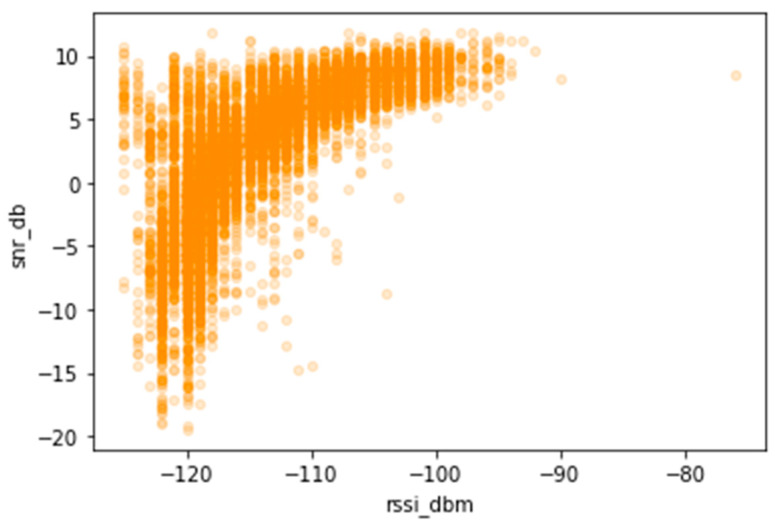
SNR as a function of RSSI.

**Figure 10 sensors-23-07394-f010:**
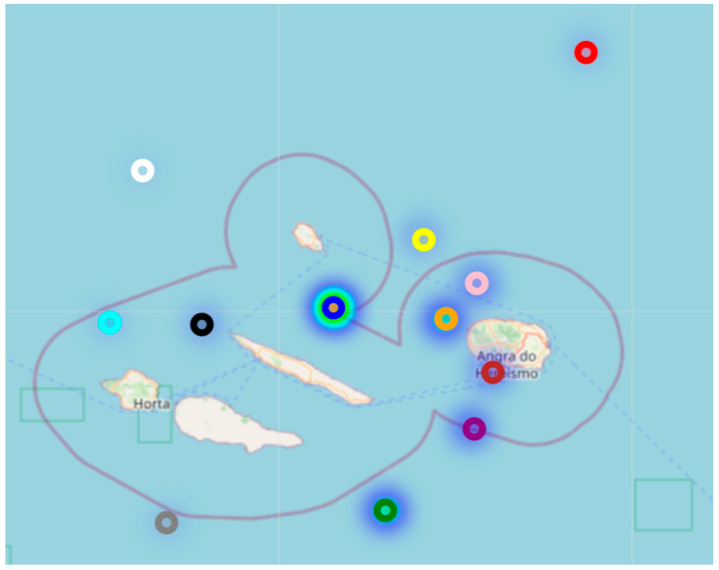
Data subsets.

**Figure 11 sensors-23-07394-f011:**
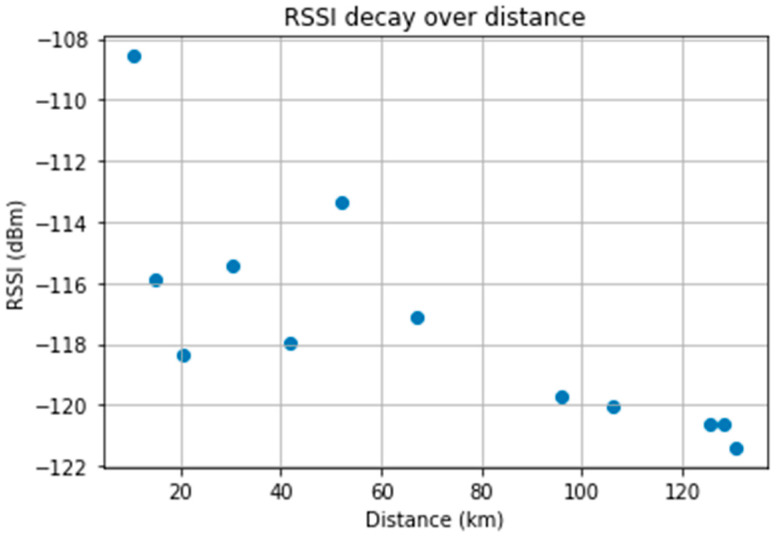
Average RSSI value of the subsets as a function of distance from the GW.

**Figure 12 sensors-23-07394-f012:**
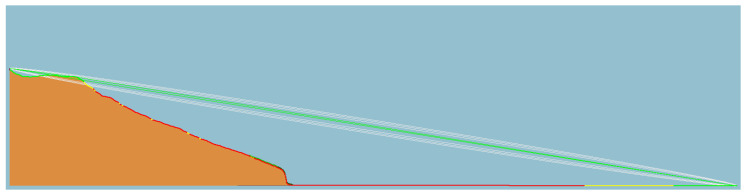
Terrain profile, line of sight, and Fresnel zone of the orange link.

**Figure 13 sensors-23-07394-f013:**
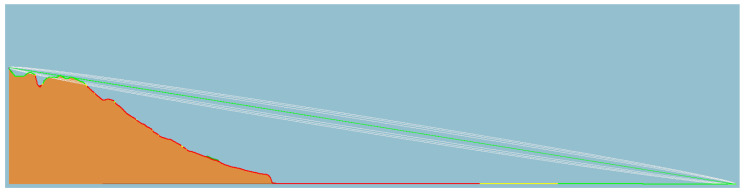
Terrain profile, line of sight, and Fresnel zone of the pink link.

**Figure 14 sensors-23-07394-f014:**
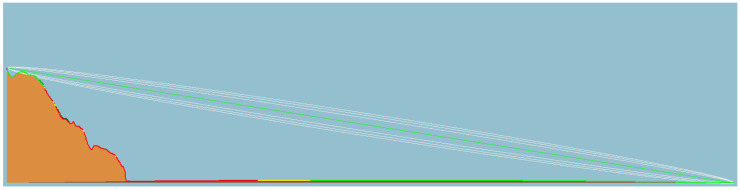
Terrain profile, line of sight, and Fresnel zone of the yellow link.

**Table 1 sensors-23-07394-t001:** LoRaWAN gateways on Terceira Island.

Location	Model	Coordinates ^1^	Altitude
Serra do Cume	Wirnet iStation	38.70945, −27.11188	563 m
Altares	Wirnet iStation	38.79919, −27.29119	156 m
Serra de Santa Bárbara	Wirnet iStation	38.73015, −27.31887	1040 m
Pico das Cruzinhas	Wirnet iStation	38.64762, −27.22546	166 m

^1^ Latitude, Longitude (WGS84).

**Table 2 sensors-23-07394-t002:** Information about the subsets.

Subset	Point Count	Center Coordinates (Latitude, Longitude)	Mean Distance to Gw (km)	Mean RSSI (dBm)	Mean SNR (dB)
Red	101	39.620395, −26.90367	106.290	−120.06	−7.37
Green	715	38.204023, −27.701205	67.26	−120.06	0.21
Blue	1654	38.834382, −27.907589	52.08	−113.34	5.25
Yellow	206	39.044477, −27.548558	41.82	−117.99	−1.63
Orange	657	38.800204, −27.459609	14.99	−115.88	1.190
Purple	475	38.458097, −27.347993	30.21	−115.42	1.62
Pink	307	38.910218, −27.337965	20.39	−118.37	−1.05
Brown	433	38.633978, −27.274188	10.63	−108.56	6.17
Gray	10	38.165462, −28.571266	125.62	−120.60	−14.16
Black	208	38.782374, −28.430655	95.75	−119.71	−4.780
White	10	39.257761, −28.666294	130.69	−121.40	−13.40
Cyan	8	38.788358, −28.796452	128.28	−120.63	−12.80

## Data Availability

The data used in this paper are available for non-commercial use at: https://github.com/AIRCentre/LoRaWAN-paper-1/tree/13b170969d3087f4c07c3b066be091438accbe34/data.

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
