# Peer review of "Unveiling LoRa’s Oceanic Reach: Assessing the Coverage of the Azores LoRaWAN Network from an Island"

_sensors, 2023, doi:10.3390/s23177394_

Round 1

Reviewer 1 Report

Authors investigate the performance of a low-power, low-cost wireless communication solution (LoRaWAN) for maritime communication of sensing data.

It is a very interesting work although there are some issues that must be fixed in order to accept the paper:

Authors should introuce the rest of the paper at the end of the introduction section.

Authors must revise what other authors have done related with the topic of the paper and cite some works like:

LoRa-based Network for Water Quality Monitoring in Coastal Areas. Mobile Networks and Applications, DOI: 10.1007/s11036-022-01994-8

Authors are using a 6dBi anthenna, while distance ranges are generally estimated based between 1 and 2 dBi. Authors should discuss why they have increased the gain and how this affects if they decrease the gain to the regular one.

The fact of having signal does not imply good communication. Authors should provide the packet loss test for these distances.

Reviewer 2 Report

This paper presents a new study, LoRa range on water. The results are interesting, on the field line of sight the coverage is about tens of km (20-30), here are much higher. From the application view, I don't think this could be the best solution (reliability, update time, packet loss), but the range tests concerning the RSSI and SNR are welcomed. The GPS can be transmitted safely in other ways, I recommend finding an application on vessels with LoRa to integrate environmental data acquisition. 
Some punctual observations are:

50: this technology has high costs and bandwidth limitations -> also LoRa has limitations, by implementing LoRa you don’t eliminate these limitations

117: sensory networks -> the term sensor networks is more used

118: The basic LoRa Network architecture -> the architecture presented is more LoRaWAN, as it is written in the figure. The description should mark a little bit more the difference between LoRa and LoRaWAN (see also line 127);

124: higher bandwidth backhauls such as 3-4-5G or Ethernet -> wi-fi should be mentioned, usually in applications before Ethernet the Wi-Fi is used

126: not sub-minute -> a detailed explanation for this is required, from LoRaWAN perspective (usually is not sub-2min)

127: must pass through the network server -> with LoRa is not mandatory, a point-to-point connection could be established (simple LoRa)

145: 2 -> it will be helpful to be detailed in a special subsection the configuration used for the transceivers (like power level, spreading factor, coding rate, bandwidth)

165: on the final version the figure name should be on the same page as the figure

224: show a discontinuity –> in this kind of application the reliability is critical; also an one minute update I don’t know if it is enough, mainly in some urgent cases (packet loss is important)

248: However -> you could delete this word, you discuss 2 different parameters there

262-270: paragraph justify (also others, please verify)

281: name of table 2 should be on the same page as the table and the values registered are better to be with one, max 2 decimals - -118.368e! )

331: range of LoRaWAN network -> here the distance is given by LoRa, LoRaWAN includes more

349: although LOS is described and is intuitive, NLOS is not defined

Some discussion about the weather is important, what is the influence of the rain for example (if there is rain), the need for such an application is for emergency and this cases usually appear on bad weather.

Round 2

Reviewer 1 Report

Authors have made all my requested changes. The paper can be published.

Author Response

The reviewer stated that: "Authors have made all my requested changes. The paper can be published."

Therefore there were no changes made.